# RETHINKING LIPSCHITZNESS DATA-FREE BACKDOOR DEFENCE

## ABSTRACT

Deep Neural Networks (DNNs) have demonstrated remarkable success across various applications, yet some studies reveal their vulnerability to backdoor attacks, where attackers manipulate models under specific conditions using triggers. It significantly compromise the model integrity. Addressing this critical security issue requires robust defence mechanisms to ensure the reliability of DNN models. However, most existing defence mechanisms heavily rely on specialized defence datasets, which are often difficult to obtain due to data privacy and security concerns. This highlights the urgent need for effective data-free defence strategies. In this work, we propose Lipschitzness Precise Pruning (LPP), a novel data-free backdoor defence algorithm that leverages the properties of Lipschitz function to detect and mitigate backdoor vulnerabilities by pruning neurons with strong backdoor correlations while fine-tuning unaffected neurons. Our approach optimizes the computation of the Lipschitz constant using dot product properties, allowing for efficient and precise identification of compromised neurons without the need of clean defence data. This method addresses the limitations of existing data-free defences and extends the scope of backdoor mitigation to include fully connected layers, ensuring comprehensive protection of DNN models. As our approach does not require data exchange, it can be implemented efficiently and effectively in diverse environments. Extensive experiments demonstrate that LPP outperforms state-of-the-art defence approaches without the need for additional defence datasets. We release our code at: https://anonymous.4open.science/r/LPP-CD3C.

## 1 INTRODUCTION

Deep Neural Networks (DNNs) have recently achieved impressive advancements in computer vision (Dhanya et al., 2022; Mahadevkar et al., 2022). For instance, DNNs outperform traditional methods on benchmark datasets for image classification tasks (He et al., 2016a; Huang et al., 2017; Sandler et al., 2018; Li, 2022; Gulzar, 2023). However, recent studies suggest that the training process of DNNs models is vulnerable to backdoor attacks (Gu et al., 2017; Chen et al., 2017). Specifically, during training, attackers can embed malicious features into the network, effectively poisoning designated neurons and creating a backdoor. When such models are subsequently used for inference on data containing stealthy implanted features, the performance will dramatically deteriorate, leading to erroneous classifications. It severely compromise the trustworthiness of DNN models. It is thus imperative to investigate robust defence mechanisms to mitigate such backdoor attacks in DNNs.

Defence against backdoor attacks can be approached from two perspectives: passive and active. Passive defence does not involve optimization of the current model but instead relies on detecting potential attack samples to provide protection, as seen in various backdoor detection methods (Dong et al., 2021; Chen et al., 2018; Liu et al., 2022). On the other hand, active defence proactively adjusts the parameters of the model to enhance its robustness and reduce the likelihood of successful attacks. Due to the significant limitations of passive defence, such as its reliance on detection algorithms and lack of real-time responsiveness, our work primarily focus on active defence mechanisms.

Active defence strategies against backdoor attacks can be categorized based on whether additional defence data is required. Both data-based and data-free methods aim to identify and either remove or modify compromised neurons. Currently, most defence mechanisms are data-based (Hinton et al., 2015; Liu et al., 2018; Li et al., 2021a; Wu & Wang, 2021; Li et al., 2023). However, these defences

rely on clean, uncontaminated samples to achieve effective performance. While such reliable data is unavailable, the effectiveness of these methods is largely reduced. To address this limitation, recent research presents data-free defence strategies, which avoid the need for clean sample during the defence process (Zheng et al., 2022a). Despite this, existing data-free methods suffer from limitations such as inappropriate matrix mappings and ineffective neuron pruning technique, leading to poor defence outcomes. Moreover, these methods are typically limited to modifying neurons in convolutional layers, neglecting potential backdoor behaviors in neurons within fully connected layers. Therefore, we introduce the concept of precise pruning to bridge these research gaps.

In this work, we introduce a novel data-free backdoor defence algorithm termed Lipschitzness Precise Pruning (LPP), as illustrated in Figure. 1. It shows the parameters which may have impacts on the decision boundary for the backdoor attacks in the models for different layers. Following the conceptual idea of CLP (Zheng et al., 2022a), we reevaluate the properties of Lipschitz Function and uncover a strong correlation between Lipschitzness and backdoor activation, which can be categorized into strong and weak correlations. By selectively removing neurons strongly associated with backdoor behaviour and fine-tuning those weakly related, LPP effectively eliminates backdoor attacks while maintaining high model performance. Additionally, we optimize the computation of Lipschitz constant using the properties of dot products, allowing for a more efficient and precise identification of backdoor neurons. This approach enables accurate detection of contaminated neurons without the need of defence data samples.

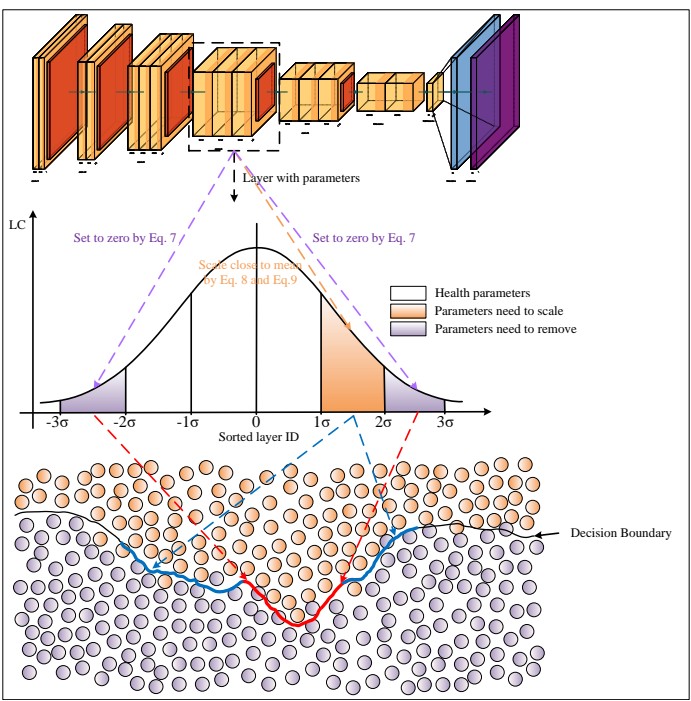

Figure 1: An illustrative diagram of LPP algorithm. We compute the corresponding Lipschitz Constants (LC) for different channels in each neural network layer, and observe the positioning of LC values within their respective distributions. Parameters exhibiting significant deviations (highlighted by purple arrows) are removed using Eq. 7, as they contribute to anomalies along the decision boundary (red arrows). Parameters with less obvious deviations (yellow arrows) are scaled using Eq. 8 and Eq. 9 to align closer to the mean, resulting in behavior similar to areas with less pronounced anomalies (blue arrows). Similar operations are also applied to the fully connected layers of the model.

We have conducted a large-scale experiment on different datasets to validate the effectiveness of our algorithm, and the results demonstrate that our approach achieves superior performance for data-free backdoor defences. Notably, compared with the state-of-the-art methods, our proposed LPP method achieves a significant performance improvement of 24.24% on average, highlighting its robust defence and generalisation capabilities without relying on clean data samples.

In summary, the contributions of this paper are as follows:

- We revisit the properties of Lipschitz functions and their equivalence with $L2$ norm, leveraging this relationship in a novel data-free defence algorithm, named Lipschitzness Precise Pruning (LPP).

- The proposed LPP method enhance the ability to precisely identify contaminated neurons based on their strong and weak correlations with backdoor behaviour.

- We conduct extensive experiments to validate the effectiveness of our approach, demonstrating 24.24% defence performance improvement in comparison to other state-of-the-art data-free defence methods.

- We release the replication package for LPP to facilitate peer review and promote future researches.

## 2 RELATED WORK

### 2.1 BACKDOOR ATTACK AND DETECTION

Backdoor attack denotes a strategy where an adversary could embed specific triggers as adversarial samples during the training of DNNs, which results in the manipulation of model behaviour. Early adversarial methods, such as BadNets (Gu et al., 2017) and Blended Attack method (Chen et al., 2017), involve introducing malicious features into the training data to implant the backdoor. Other methods, i.e., the Input-Aware Backdoor Attack (Nguyen & Tran, 2020) method, alternatively selects a set of data with specific patterns, characteristics, or attributes to implant a more concealed backdoor, rendering it less conspicuous. Furthermore, altering the data via specific transformations, such as translation, rotation, scaling, noise addition, color alteration and so on in Warping-based Backdoor Attack method (Nguyen & Tran, 2021), make the backdoor activation dependent on manipulated distorted data, thus enhancing the secrecy of the attack. Another approach is the SIG method (Barni et al., 2019), which targets backdoor attacks on specific labels by injecting triggers into samples of the target label. During testing, if the input sample contains the trigger, the model is misled into classifying it as the designated label.

Unlike previous attack methods that require backdoor labeling for a set of data, the Sample Specific Backdoor Attack (Li et al., 2021b) only use a single sample, rendering the backdoor even more challenging to detect. Additionally, the BPP Attack (Wang et al., 2022) employs a multi-step process that first quantizes and perturbs images to generate backdoor triggers. It then employs contrastive learning and adversarial training to contaminate the DNN model, thereby enhancing both the stealth and effectiveness of the attack.

To counter these evolving threats, detection mechanisms like Black-box Backdoor Detection (B3D) (Dong et al., 2021) offer a strategy that works under black-box conditions, requiring neither internal model access nor tainted data. By employing a gradient-free optimization approach, B3D refines potential trigger characteristics, identifying backdoors through output discrepancies. Activation Clustering (AC) (Liu et al., 2022) focuses on detecting uniform activation patterns triggered by backdoors, using hidden layers to identify poisoned inputs. Similarly, EX-RAY (Liu et al., 2022) scrutinizes feature maps for backdoor-related anomalies by detecting irregularities in symmetry. However, these passive defense methods are limited by their dependence on detection algorithms, significant computational overhead, and inability to proactively prevent attacks. They mitigate impacts post-attack but struggle with novel or complex backdoor methods, making proactive solutions essential for robust defense.

### 2.2 DATA-BASED BACKDOOR DEFENCE

To address the growing threat of backdoor attacks, numerous defence strategies have been proposed in two categories: data-based and data-free methods. A significant portion of the research focuses on data-based approaches, starting with fine-tuning (FT) (Hinton et al., 2015), which adjusts parameters in a pre-trained model to reduce or eliminate the effects of backdoor triggers. Expanding on this, Fine-Pruning (FP) (Liu et al., 2018) combines network pruning and fine-tuning to remove redundant structures and weaken backdoor influences. Neural Attention Distillation (NAD) (Li et al., 2021a)

further enhances fine-tuning by incorporating knowledge distillation, guiding a contaminated model using a clean teacher model to align its intermediate layer attention.

Several pruning-based defences have also emerged, such as Adversarial Neuron Pruning (ANP) (Wu & Wang, 2021), which exploits the sensitivity of backdoor-affected neurons by pruning those linked to adversarial triggers, effectively neutralizing the backdoor while preserving model performance. BN statistics-based pruning (BNP) (Zheng et al., 2022b) relies on discrepancies in Batch Normalization statistics to identify and prune contaminated neurons. Similarly, Reconstructive Neuron Pruning (RNP) (Li et al., 2023) uses a forgetting-recovery process to retrain a backdoored model by identifying and removing compromised neurons. Another one, Implicit Backdoor Adversarial Unlearning (I-BAU) (Zeng et al., 2021), minimizes backdoor effects by jointly optimizing contaminated and clean models, using implicit gradients to enhance robustness.

These defence mechanisms, particularly those involving pruning and fine-tuning, demonstrate an evolving effort to mitigate backdoor attacks, focusing on improving detection and eliminating compromised components from DNNs without significantly degrading performance.

### 2.3 DATA-FREE BACKDOOR DEFENCE

In contrast to the extensive research on data-based backdoor defence methods, data-free backdoor defence is still in its early stages. Despite this, data-free approaches hold significant potential in addressing practical challenges, such as the difficulty in obtaining large amounts of clean data due to cost, security, and privacy concerns. One recent method is Lipschitzness-based Pruning (CLP) (Zheng et al., 2022a), which assesses the contribution of each channel in the neural network by calculating the Lipschitz constant and removes channels with values below a certain threshold. Since the Lipschitz constant can be computed directly from model parameters, CLP eliminates the need for clean data during defence. However, this channel-level pruning lacks precision in targeting contaminated neurons.

To improve the granularity of this approach, we introduce the concept of precise pruning, offering a more accurate means of trimming compromised neurons while maintaining the model's performance. Precise pruning provides finer control over which neurons are targeted, enhancing the effectiveness of backdoor defences by addressing contamination without the need for clean datasets, thus offering a valuable solution to the limitations of current data-free strategies.

## 3 METHOD

### 3.1 PRELIMINARIES

#### 3.1.1 PROBLEM DEFINITION

In Equation. 1, it represents the definition of a backdoor attack. Here, $\theta$ denotes the model parameters, $\mathbb{E}$ represents the expectation operator, $\mathcal{L}$ denotes the loss function, and $f$ signifies the model. $\boldsymbol{x}_c$ and $y_c$ denote clean data and their respective labels, while $\boldsymbol{x}_b$ and $y_b$ represent backdoor data and their corresponding labels. Notably, $f(\boldsymbol{x}_c, y_c; \theta)$ pertains to the clean task, while $f(\boldsymbol{x}_b, y_b; \theta)$ pertains to the backdoor task.

$$\min_{\theta} \left[ \mathbb{E}_{\substack{(\boldsymbol{x}_c, y_c) \in \mathcal{D}_c \\ (\boldsymbol{x}_b, y_b) \in \mathcal{D}_b}} \left[ \underbrace{\mathcal{L}(f(\boldsymbol{x}_c, y_c; \theta))}_{\text{Clean Task}} + \underbrace{\mathcal{L}(f(\boldsymbol{x}_b, y_b; \theta))}_{\text{Backdoor Task}} \right] \right] \quad (1)$$

The entire optimization process of the backdoor attack aims to concurrently identify the model parameters $\theta$ in order to achieve outstanding performance on both the clean and backdoor tasks. Conversely, the objective of backdoor defence is to find the model parameters $\theta$ such that, without compromising the performance of the clean task, the performance of the backdoor task is minimised to the greatest extent possible.

### 3.1.2 BATCH NORMALIZATION

$$\left(x_j^l\right)' = \gamma_j \left(\frac{x_j^l - \mu_j^l}{\sqrt{(\sigma_j^l)^2 + \epsilon}}\right) + \beta_j \tag{2}$$

where $x_j^l$ represents the input features of layer $l$, $\mu_j^l$, $\sigma_j^l$, $\gamma_j$, and $\beta_j$ denote the mean, standard deviation, scale parameter, and bias parameter of $j$-th channel and $l$-th layer, respectively.

### 3.2 RETHINKING LIPSCHITZ FUNCTION

$$||f(x_1) - f(x_2)||_p \leq C||x_1 - x_2||_p \tag{3}$$

As depicted in Equation. 3, a function that satisfies the condition for all $x_1$ and $x_2$, where $C$ is a constant independent of $x_1$ and $x_2$, is commonly referred to a Lipschitz function (LF) (Armijo, 1966). If we interpret $||f(x_1) - f(x_2)||_p$ as $\Delta y$ and consider $||x_1 - x_2||_p$ as $\Delta x$, we can perceive $C$ as the maximum gradient value $\Delta y/\Delta x$. The magnitude of $C$ directly reflects the degree of abruptness in the function's variation. A larger value of $C$ indicates a greater upper bound on the gradient of the function, which in turn implies that the function $f$ is more unstable under worst-case scenarios.

$$f^{(l)}(x) = \begin{cases} w^l x + b^{(l)} \\ \sigma(x) \end{cases} \tag{4}$$

We consider the neural network in numerous layers, and here we denote the transformation function of the $l$-th layer of the neural network as represented in Equation. 4, where $w^l$ and $b^l$ denote the model parameters of the $l$-th layer, $x$ represents the input features and $\sigma$ represents the activation function. Since the activation function lacks trainable parameters, it falls outside the focus of the pruning methods we are exploring. However, it's important to note that convolution functions can be viewed as sparsely connected fully connected neural networks with shared weights and can be expressed using the same mathematical formulas as fully connected networks. In the following discussion, we employ $||f^l||_{lip}$ to denote the Lipschitz constant (LC) of the function.

$$F(x) = \left(f^{(l)} \circ f^{(l-1)} \circ \cdots \circ f^{(1)}\right)(x) \tag{5}$$

$$\begin{aligned} ||F||_{lip} &= \left\|f^{(l)} \circ f^{(l-1)} \circ \cdots \circ f^{(1)}\right\|_{lip} \\ &\leq \left\|f^{(l)}\right\|_{lip} \cdot \left\|f^{(l-1)}\right\|_{lip} \cdots \left\|f^{(1)}\right\|_{lip} \end{aligned} \tag{6}$$

As illustrated in Equation. 5, we conceive the neural network as a parallel composition of multiple functions and employ the Lipschitz Function to monitor the operational process of the neural network. Given that the LC characterizes the maximum extent of change a layer can induce in its variables, the overall model's variation must be bounded by the cumulative product of the LCs of each layer. This relationship is formally expressed as per Equation 6.

Inspired by CLP, our approach to neuron pruning is grounded in the utilization of Lipschitz Functions. In CLP, Zheng et al. (Zheng et al., 2022a) employ Lipschitz Functions to assess the importance of different channels within convolutional kernels, thereby reducing the model's backdoor behavior by pruning specific channels of the kernels. We define the input dimension of layer $f$ as $c_2$, and the output dimension as $c_1$. CLP leverages Lipschitz Functions by treating the convolutional kernel $W \in \mathbb{R}^{c_1 * k * k * c_2}$, as a collection of $c_1$ kernels $W_j \in \mathbb{R}^{k * k * c_2}$, and $j \in \{0, 1, 2, \cdots, c_1\}$. Additionally, each kernel is reshaped into a doubly block-Toeplitz (DBT) form in the matrix space

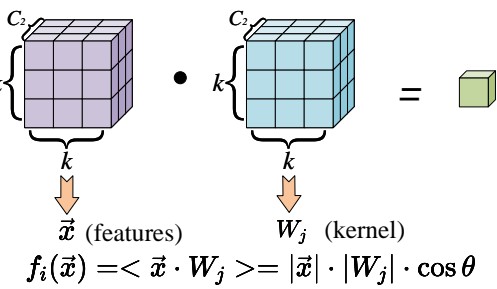

$$f_i(\vec{x}) = <\vec{x} \cdot W_j> = |\vec{x}| \cdot |W_j| \cdot \cos\theta$$

Figure 2: Illustration of dot products

$\mathbb{R}^{c_2 \times (k * k)}$. Through singular value decomposition (SVD) of this matrix and observation of the distribution of the largest eigenvalues among all $c_1$ kernels, if they exceed $u$, the hyperparameter of

LPP, standard deviations from the mean of these kernels, the entire channel is removed. However, CLP presents two issues. **Issue 1:** as it necessitates the removal of entire channels, it is not applicable to fully connected neural network layers lacking channels, which also leads to the inability to precisely locate neurons for removal. **Issue 2:** The convolutional kernel of each channel can be regarded as a transformation matrix, which maps features from a dimension of k*k to c2. However, in actual convolutional kernel operations, **the functionality of the kernel** can be viewed as a transformation matrix that maps features from dimensions $k * k * c_2$ to dimension 1, resulting in a discrepancy between the eigenvalues obtained from singular value decomposition and the actual functionality of the transformation matrix.

Based on our prior discussion on the correct functionality of convolutional kernels, we can view it as a dot product relationship like Figure 2 between the kernel $W_i \in \mathbb{R}^{k*k*c_2}$ and the input vector $x$ in $\mathbb{R}^{k*k*c_2}$, denoted as $f_j(\vec{x}) = <\vec{x} \cdot W_j> = |\vec{x}| \cdot |W_j| \cdot \cos\theta$. This allows us to utilize the properties of dot products to estimate the Lipschitz Constant (LC) of $W_j$. Under the Data-free constraint, the magnitudes of $\vec{x}$ and $\cos\theta$ remain unknown in $f_j(\vec{x})$, and the rate of function variation is solely dependent on $W_j$. LC can be assessed using the norm of vector $W_j$. Therefore, we can employ $W_j$ to evaluate the convolutional kernels across different channels. Now, we can calculate $|W_j|$ to compute the LC value for the $j$-th channel of the $l$-th layer, denoted as $LC_j^l$. This approach helps circumvent **Issue 2** encountered in CLP. It is noteworthy that squaring does not alter the relative magnitude relationships of the Lipschitz Constants (LCs). Therefore, we can simplify the computation using $||W_j||_2$. This is because, in the mapping process, the functional role of the convolutional kernel as a transformation matrix remains unchanged, and consequently, the corresponding mapped space also remains unaltered. Additionally, as we utilize $W_j$ to assess the LC, the computation process of each neuron's parameters is independent, allowing for the individual evaluation of each neuron's impact on the LC, thereby mitigating **Issue 1** observed in CLP.

### 3.3 LIPSCHITZNESS PRECISE PRUNING

As shown in Figure. 3, we observe a clear positive correlation between the Lipschitz constant (LC) and the incidence of backdoor triggers. Therefore, LC can serve as a basis for pruning neuron parameters to mitigate backdoor behavior. Our experimental design is based on 100 clean samples from the Tiny ImageNet dataset and their corresponding backdoor samples generated by the BPP attack method. The applied model is ResNet18, and no specific defense mechanisms were applied within the model to ensure that the results were not affected by additional variables. Figure 3 illustrates the correlation between the differences in the feature map outputs of these clean samples and their corresponding backdoor samples, and the Lipschitz constants (LC). We chose to use the correlation obtained through `np.corrcoef` between the difference in the output of features of the neurons before and after adding backdoor features and the LC calculated for that neuron. It was found that there are two peaks in the correlation between LC and backdoor behavior.

Before proceeding with precise pruning, we firstly analyse the properties of the dot product. The dot product can be seen as the element-

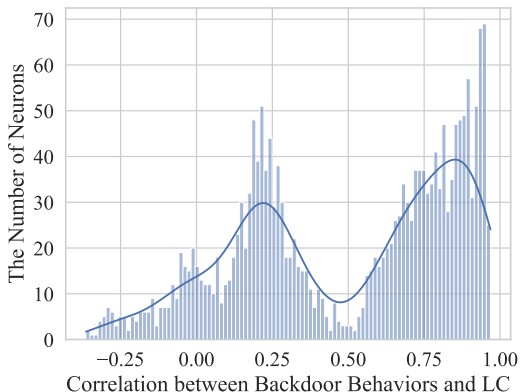

Figure 3: Correlation between the output difference with and without a backdoor trigger and the Lipschitz Constant (LC). Two distinct peaks are visible—indicating weak correlation on the left and strong correlation on the right. Neurons with strong correlation are pruned, while weakly correlated ones are scaled to reduce the Lipschitz constants.

wise multiplication followed by summation of corresponding values in the vectors $\vec{x}$ and $\vec{u}$. Furthermore, in the Lipschitz Function (LF), the representation of the upper bound of the LC is solely related to $W_j$. We can consider dimensions in the $W_j$ vector with higher values as potential locations where backdoor neurons may exist. This is because, once the corresponding value of $\vec{x}$ increases

in a dimension, the result of the dot product will sharply rise, thereby meeting the trigger condition for the backdoor behavior. Hence, it is likely that backdoor neurons reside in **dimensions of** $W_j$ **with larger values**. Additionally, concerning the specific scenario immediately preceding the Batch Normalization (BN) layer, the value of LC can be adjusted by BN layer parameters as follows: $LC_j^l = \frac{LC_j^l}{r_j^l} \cdot \sigma_j^l$. This adjustment stems from the interpretation of the BN and the transformation in the preceding layer as a composite function.

Our LPP algorithm consists of two components: the removal of severely biased parameter channels and the application of scaling to parameters exhibiting bias.

$$P_{idx} = \{\{l, j\} : LC_j^l > \mu^l + u * s^l\} \cup \{\{l, j\} : LC_j^l < \mu^l - u * s^l\} \tag{7}$$

Where $\mu^l$ denotes the mean of the LC across all channels in the $l$-th layer, and $s^l$ represents the standard deviation of the LC across all channels in the same layer. As illustrated in Equation 7, we identify channels with significantly large deviations in LC values and subsequently set all output values of these channels to zero. The blue portion in Figure 2 corresponds to these severely biased parameters.

$$S_{idx} = \{\{l, j\} : LC_j^l > \mu^l + (u - b) * s^l\} \cap \{\{l, j\} : LC_j^l < \mu^l + u * s^l\} \tag{8}$$

$$W_j^l = W_j^l \times \frac{\mu^l}{LC_j} \quad \{l, j\} \in S_{idx} \tag{9}$$

Subsequently, as per Equation. 8, $b$ represents the bias rate, we identify channels exhibiting bias, following we apply an adjustment to selected channels using Equation 9 to bring them closer to an unbiased state.

## 4 EXPERIMENT

In this section, we discuss our comprehensive evaluation, including the setup, metrics, and key results. We begin by presenting our results based on the experimental design from CLP to facilitate a fair comparison. Additional experiments, such as extended model evaluations and detailed performance analysis, computational efficiency assessments, and an ablation study, are included in the Appendix C.

### 4.1 EXPERIMENTAL SETUP

#### 4.1.1 DATASET

To ensure a fair comparison, we utilized the same dataset as CLP. We conducted experiments on CIFAR-10 (Krizhevsky et al., 2009) and Tiny ImageNet (Le & Yang, 2015). Additionally, we introduced the German Traffic Sign Recognition Benchmark (GTSRB) (Houben et al., 2013) for further comparison, thus validating the effectiveness of our approach. We employed 1% of the training data as benign data for the Data-based defence algorithm.

#### 4.1.2 MODEL TRAINING SETUP

We trained the aforementioned datasets on the ResNet-18 (He et al., 2016b) model. For both training CIFAR-10, Tiny ImageNet and GTSRB, the batch size was set to 128, momentum was set at 0.9, and the base optimizer used was SGD. There were slight variations in the learning rate, epoch, and adjust the learning rate strategy. Specifically, the learning rate for CIFAR-10 was set at 0.001, while for Tiny ImageNet and GTSRB, it was set at 0.01. The epochs were 100 for CIFAR-10, 50 for Tiny ImageNet, and 200 for GTSRB. CIFAR-10 and Tiny ImageNet employed the Cosine scheduler, whereas GTSRB utilized the Reduce learning rate scheduler.

#### 4.1.3 BACKDOOR ATTACK SETUP

In this experiment, we employed four representative Backdoor attack algorithms, namely Bad-Net (Gu et al., 2017), BPP (Wang et al., 2022), Inputaware (Nguyen & Tran, 2020), and WaNet (Nguyen & Tran, 2021). To maintain experimental fairness, all Backdoor attack methods

in this study followed the settings of our primary competing algorithm CLP (Zheng et al., 2022a). Specifically, the training approach for the attack categories utilized the All-to-One strategy, wherein all samples of the attacked data were labeled as the same category. All attack methods set the first label as the contamination label. The contamination rates on CIFAR-10 and GTSRB were set at 10% and 1%, respectively, while on Tiny ImageNet, it was set at 10%. The Trigger size across all datasets was uniformly set at 3x3.

We establish the comparative methods against the state-of-the-art approaches including data-based and data-free backdoor defence methods, including FT, FP (Liu et al., 2018), NAD (Li et al., 2021a), ANP (Wu & Wang, 2021), I-BAU (Zeng et al., 2021), the SOTA neuron pruning strategy BNP (Zheng et al., 2022b), and the SOTA data-free method CLP (Zheng et al., 2022a). Due to the page limit, the full results can be obtained in the replication package. In following sections, we focus on the discussion among the results from the methods of BNP, I-BAU, NAD and CLP.

## 4.2 EVALUATION METRIC

In this experimental study, we adhered to the evaluation metrics proposed by CLP (Zheng et al., 2022a), employing both Accuracy on Clean data (ACC) and Attack Success Rate (ASR) to assess the performance of Backdoor defence algorithms. ACC represents the accuracy achieved on normal, uncontaminated data, while ASR quantifies the proportion of successfully attacked instances among the contaminated data. Consequently, a higher ACC coupled with a lower ASR signifies enhanced defence performance of the algorithm.

## 4.3 EXPERIMENTAL RESULTS

Table 1: Performance Comparison of Defence Methods on CIFAR-10 Dataset. The greater the disparity between ACC and ASR, the more effectively the defence method has accomplished its purpose, namely, to maintain high ACC while reducing ASR. Therefore, we have highlighted in bold the data with the maximum disparity between ACC and ASR.

| Poison Data Rate | Attack Method | No Defence | | Data-based Defence | | | | | | Data-free Defence | | | |
|---|---|---|---|---|---|---|---|---|---|---|---|---|---|
| | | | | BNP | | I-BAU | | NAD | | CLP | | LPP | |
| | | ACC | ASR | ACC | ASR | ACC | ASR | ACC | ASR | ACC | ASR | ACC | ASR |
| 10% | BadNet | 88.79 | 94.96 | 88.3 | 95.29 | 14.8 | 4.63 | 30.97 | 64.31 | 84.82 | 3.97 | **85.51** | **2.81** |
| | BPP | 90.34 | 99.48 | 90.07 | 3.37 | 14.5 | 99.77 | 78.05 | 10.33 | 90.31 | 2.86 | **90.32** | **2.791** |
| | Inputaware | 89.66 | 97.61 | 90.08 | 3.52 | 89.52 | 67.93 | 91.81 | 93.66 | 89.62 | 2.2 | **90.27** | **0.68** |
| | WaNet | 86.06 | 99.25 | 57.65 | 97.41 | 25.93 | 2.56 | **81.76** | **1.88** | 68.97 | 96.22 | 87.64 | 44.57 |
| 1% | BadNet | 93.56 | 77.14 | 93.59 | 76 | 74.12 | 9.98 | 93.63 | 69.71 | 91.24 | 9.04 | **92.1** | **4.23** |
| | BPP | 91.45 | 85.95 | **88.12** | **1.77** | 90.72 | 91.61 | 92.79 | 98.19 | 67.31 | 91.42 | 91.17 | 3.225 |
| | Inputaware | 89.7 | 79.92 | 91.54 | 86.61 | 87.47 | 69.18 | 92.14 | 91.6 | 90.83 | 0.97 | **91.21** | **0.6** |
| | WaNet | 91.06 | 51.66 | 56.83 | 88.84 | 88.92 | 3.96 | 92.63 | 19.57 | 90.2 | 0.88 | **90.02** | **0.82** |

### 4.3.1 EFFECTIVENESS ANALYSIS

In this experiment, our LPP method demonstrated superior performance in most cases. As illustrated in Table. 1, when considering ACC and ASR jointly on the CIFAR-10 dataset, our approach exhibited improvements for almost all attack methods. It can largely maintain the model prediction performance as for the No Defence scenario while minimising the ASR values.

Experimental results on the CIFAR-10 dataset revealed a significant advantage of the LPP method over other defence approaches in terms of reducing the ASR. Overall, in comparison to the absence of defence, the LPP method experienced a mere 0.238% reduction in ACC, while achieving an average increase of approximately 62.62%, with a maximum improvement of up to 96.69% in ASR. This signifies a notable advantage of the LPP method in diminishing the success rate of backdoor attacks.

Relative to data-based methodologies, the LPP method exhibited an average increase in ACC of 11.96%, with a maximum improvement of 75.82%. Regarding ASR, our method demonstrated an average reduction of 35.75%, with a maximum decrease of 96.98%. In comparison to our primary competitor, the Data-free CLP method, the LPP method demonstrated an average increase of 4.49% in ACC, and in terms of ASR, it exhibited an average increase of 14.78%.

Table 2: Performance Comparison of Defence Methods on GTSRB Dataset. The greater the disparity between ACC and ASR, the more effectively the defence method has accomplished its purpose, namely, to maintain high ACC while reducing ASR. Therefore, we have highlighted in bold the data with the maximum disparity between ACC and ASR.

| Poison Data Rate | Attack Method | No Defence | | Data-based Defence | | | | | | Data-free Defence | | | |
| --- | --- | --- | --- | --- | --- | --- | --- | --- | --- | --- | --- | --- | --- |
| | | | | BNP | | I-BAU | | NAD | | CLP | | LPP | |
| | | ACC | ASR | ACC | ASR | ACC | ASR | ACC | ASR | ACC | ASR | ACC | ASR |
| 10% | BadNet | 96.87 | 95.02 | 97.04 | 94.51 | 78.73 | 10.64 | 97.97 | 82.83 | 96.81 | 58.87 | **96.42** | **1.51** |
| | BPP | 97.95 | 99.95 | 97.97 | 85.82 | 87.16 | 0 | 98.31 | 1.15 | 97.82 | 6.07 | **98.09** | **0.02** |
| | Inputaware | 98.03 | 92.21 | **98.17** | **0.2** | 93.04 | 2.72 | 98.37 | 99.62 | 98.05 | 59.45 | 95.57 | 8.41 |
| | WaNet | 97.84 | 97.65 | 6.14 | 100 | 94.96 | 33.79 | 98.92 | 44.34 | 44.09 | 99.96 | **98.53** | **13.31** |
| 1% | BadNet | 98.25 | 89.37 | 98.02 | 88.99 | 8.71 | 0 | 98.37 | 86.87 | 98.18 | 86.91 | **95.51** | **7.91** |
| | BPP | 98.02 | 59.21 | 97.93 | 0.16 | 95.38 | 0.33 | 98.5 | 74.72 | 98.04 | 12.25 | **98.27** | **0.02** |
| | Inputaware | 98.32 | 27.35 | 98.41 | 17.03 | 92.86 | 6.02 | 98.81 | 20.91 | 98.55 | 0.61 | **97.89** | **0.02** |
| | WaNet | 97.97 | 35.96 | 97.22 | 33.52 | 96.34 | 25.21 | 98.91 | 31.46 | 97.75 | 31.53 | **96.62** | **7.12** |

Table 3: Performance Comparison of Defence Methods on Tiny ImageNet Dataset. The greater the disparity between ACC and ASR, the more effectively the defence method has accomplished its purpose, namely, to maintain high ACC while reducing ASR. Therefore, we have highlighted in bold the data with the maximum disparity between ACC and ASR.

| Benign Data Rate | Attack Method | No Defence | | Data-based Defence | | | | | | Data-free Defence | | | |
| --- | --- | --- | --- | --- | --- | --- | --- | --- | --- | --- | --- | --- | --- |
| | | | | BNP | | I-BAU | | NAD | | CLP | | LPP | |
| | | ACC | ASR | ACC | ASR | ACC | ASR | ACC | ASR | ACC | ASR | ACC | ASR |
| 5% | BadNet | 59.48 | 99.91 | 59.48 | 99.9 | 53.87 | 92.43 | 50.69 | 0.97 | 59.36 | 90.84 | **58.4** | **0.75** |
| | BPP | 61.25 | 100 | 61.01 | 99.98 | 56.75 | 1.43 | 49.39 | 0.31 | 60.6 | 0.21 | **60.88** | **0.07** |
| | Inputaware | 61.37 | 99.61 | 32.16 | 96.67 | 56.83 | 6.15 | 51.42 | 0.11 | 61.21 | 15.97 | **60.95** | **1.5** |
| | WaNet | 61.13 | 99.93 | 60.88 | 99.89 | 55.12 | 93.54 | 48.8 | 1.19 | 61.25 | 18.54 | **59.62** | **0.29** |
| 1% | BadNet | 59.48 | 99.91 | 59.48 | 99.91 | 49.08 | 87.02 | 57.07 | 97.77 | 59.36 | 90.84 | **58.4** | **0.75** |
| | BPP | 61.25 | 100 | 61.01 | 99.99 | 48.3 | 76.67 | 59.18 | 0.54 | 60.6 | 0.21 | **60.88** | **0.07** |
| | Inputaware | 61.37 | 99.61 | 32.16 | 96.67 | 49.16 | 18.61 | 60.67 | 39.42 | 61.21 | 15.97 | **60.95** | **1.5** |
| | WaNet | 61.13 | 99.93 | 60.88 | 99.9 | 50.47 | 98.24 | 57.88 | 0.92 | 61.25 | 18.54 | **59.62** | **0.29** |

On the GTSRB dataset, in the majority of attack scenarios, our LPP method has demonstrated more reasonable levels of both ACC and ASR compared to other defence techniques. This signifies the endeavor to maintain high ACC while minimizing ASR under the precondition of achieving the lowest possible ASR. For instance, as shown in Table 2, in the case of a BadNet attack, although the NAD defence method achieved the highest ACC, its ASR reached as high as 82.83%, indicating a fundamental inadequacy in thwarting attacks from backdoor samples. In contrast, relative to NAD, our approach successfully reduced ASR by 81.32% with a modest loss of only 1.55% in ACC. Furthermore, we observed that Data-based Defence methods exhibited significant performance fluctuations when the Poison Data Rate was low, whereas Data-free defence methods displayed greater stability. When attackers employ a strategy involving minimal data contamination, this more covert form of attack is better suited for defence using Data-free methods.

For the more complicated Tiny ImageNet dataset, as shown in Tabel 3, our LPP method demonstrated the most advanced performance. Overall, in comparison to scenarios without any defensive measures, LPP exhibited a modest average reduction of 0.68% in ACC, while achieving a substantial improvement in ASR, with an average increase of 79.37%. In the meantime, we also observed that certain defence mechanisms exhibited diminished defensive efficacy on complex datasets. For example, BNP faltered in its defensive capabilities against all attack methods, and CLP also lost its defence effectiveness against certain attack methods. In contrast, LPP exhibited robust defensive performance across all attack scenarios.

## 5 CONCLUSION

In this paper, we address the critical issue of backdoor attacks on DNNs. We propose a novel data-free defence mechanism, named Lipschitzness Precise Pruning (LPP), which improves the backdoor defence of DNN models without the need of clean defence datasets and extensive computa-

tional resources such as GPU. By rethinking the Lipschitzness continuous property and devising a precise pruning approach, we efficiently eliminate the tainted channels and precisely identify the neurons contributing to backdoor attacks. Our extensive experiments validate the state-of-the-art performance of LPP method, demonstrating substantially improved results on different datasets. We anticipate our work can contribute a practical and efficient defence mechanism against backdoor attacks, while simultaneously addressing the limitations of existing defence methods, especially in situations where access to clean data is limited. We believe that LPP method has great potential for safeguarding the trustworthiness of deep neural networks in real-world applications.

## CODE OF ETHICS AND ETHICS STATEMENT

All authors of this paper have adhered to the ICLR Code of Ethics, as outlined at `https://iclr.cc/public/CodeOfEthics`. We have thoroughly reviewed and followed the ethical guidelines during all phases of research, from the inception of the project to the submission of this paper. Our study does not involve human subjects, and the datasets used in our experiments are publicly available and anonymized, ensuring the protection of privacy and data security. We acknowledge no potential conflicts of interest, sponsorship, or legal compliance issues related to this research. Furthermore, we have ensured that our research practices uphold the highest standards of integrity, including full transparency of our methods and results.

## REPRODUCIBILITY STATEMENT

To ensure the reproducibility of our results, we have provided detailed descriptions of the methods and experimental setups in the main text and supplementary materials. Our novel Lipschitzness Precise Pruning (LPP) algorithm, as well as the experimental setups for the datasets and models, are comprehensively documented. Furthermore, the code for the LPP algorithm and the data processing steps for the datasets (CIFAR-10, Tiny ImageNet, and GTSRB) used in our experiments will be made available in the anonymous supplementary materials. A clear explanation of the theoretical assumptions and all necessary proofs are included in the appendix to facilitate replication of our results.

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

## A  VARIABLES AND SYMBOLS

| Symbol | Meaning |
|---|---|
| $\theta$ | Model parameters |
| $\mathbb{E}$ | Expectation operator |
| $\mathcal{L}$ | Loss function |
| $f$ | Model function |
| $\boldsymbol{x}_c, y_c$ | Clean data and their corresponding labels |
| $\boldsymbol{x}_b, y_b$ | Backdoor data and their corresponding labels |
| $\mathcal{D}_c, \mathcal{D}_b$ | Clean dataset and backdoor dataset |
| $x_j^l$ | Input features for the $l^{th}$ layer |
| $\mu_j^l, \sigma_j^l$ | Mean and standard deviation for the $j^{th}$ channel and $l^{th}$ layer |
| $\gamma_j, \beta_j$ | Scale and bias parameters for the $j^{th}$ channel |
| $C$ | Constant, independent of $x_1$ and $x_2$ |
| $\Delta y, \Delta x$ | Change in the function value |
| $w^l, b^l$ | Model parameters for the $l^{th}$ layer |
| $\sigma$ | Activation function |
| $F(x)$ | Composition of multi-layered functions |
| $\|F\|_{lip}$ | Lipschitz constant of the model |
| $LC_j^l$ | Lipschitz constant for the $j^{th}$ channel of the $l^{th}$ layer |
| $P_{idx}$ | Indices of severely biased parameter channels that need to be removed |
| $S_{idx}$ | Indices of biased parameter channels that need adjustment |
| $u, b$ | Hyperparameters for determining the bounds of the Lipschitz constant |
| $\mu^l, s^l$ | Mean and standard deviation of Lipschitz constants for the $l^{th}$ layer |

## B  PSEUDOCODE

---
**Algorithm 1** Lipschitzness Precise Pruning
---
**Input:** Parameter Matrix $W$, The Degree of Bias $u$, The Extreme Parameter Number $k$, The Bias Rate $b$
**Output:** $W$
1: **Initial:** $b = 1.5$ (Fixed parameters)
2: **for** $l = 0, 1, \cdots, L-1$ **do**
3:  **if** layer is convolutional layer **then**
4:   **for** $j = 0, 1, \cdots, c-1$ **do**
5:    $LC_j^l = \|W_j^l\|_2$
6:   **end for**
7:   $\mu^l = \frac{1}{c} \sum_{j=0}^{c-1} LC_j^l$
8:   $s^l = \sqrt{\frac{1}{c} \sum_{j=0}^{c-1} (LC_j^l - \mu^l)^2}$
9:   $P_{idx} = \{\{l, j\} : LC_j^l > \mu^l + u * s^l\} \cup \{\{l, j\} : LC_j^l < \mu^l - u * s^l\}$
10:   pruning $W^l$ by $P_{idx}$
11:   $S_{idx} = \{\{l, j\} : LC_j^l > \mu^l + (u - b) * s^l\} \cap \{\{l, j\} : LC_j^l < \mu^l + u * s^l\}$
12:   Scaled $W^l$ by $S_{idx}$ with Equation 9
13:  **end if**
14:  **if** layer is convolutional layer **then**
15:   $maxId = \arg\max_{i} mean(W[i, :])$
16:   $T_{idx} = top_k(W[maxId, :] - mean(W[i \neq maxId, :], 0))$
17:   remove $top_k$ parameters in $W^l$
18:  **end if**
19: **end for**
20: **return** $W$

---

## C  ADDITIONAL EXPERIMENTAL RESULT

### C.1  EXPERIMENTAL RESULT ON VGG19

We have added experiments using the VGG19 model in the CIFAR-10 dataset, with the benign data rate set to 10%. The experimental results are shown in the table below. It can be observed that our

LPP method outperforms the CLP defense mechanism, which is also Data-free, in terms of both ACC (Accuracy) and ASR (Attack Success Rate). Moreover, compared to other data-based defense mechanisms, our approach demonstrates superior results.

| | BadNet | | BPP | | Inputaware | | WaNet | |
|---|---|---|---|---|---|---|---|---|
| | ACC | ASR | ACC | ASR | ACC | ASR | ACC | ASR |
| No Defence | 90.40% | 94.71% | 95.49% | 4.28% | 93.49% | 5.51% | 96.48% | 3.17% |
| BNP | 89.80% | 95.01% | 89.63% | 96.24% | 89.41% | 2.52% | 54.55% | 98.31% |
| I-BAU | 81.91% | 0.68% | 88.57% | 76.27% | 87.55% | 52.84% | 89.16% | 1.48% |
| NAD | 83.24% | 51.91% | 89.18% | 3.02% | 89.61% | 28.68% | 90.33% | 38.12% |
| CLP | 85.93% | 8.41% | 89.30% | 3.23% | 89.78% | 9.29% | 79.25% | 3.48% |
| LPP | 87.53% | 6.90% | 89.41% | 1.83% | 89.57% | 2.04% | 84.44% | 3.47% |

## C.2 IMPACT OF LPP DEFENSE ON CLEAN MODELS

We conducted additional evaluations on the clean ResNet18 model to assess the potential impact of our LPP defense method on the classification accuracy of clean models. After applying our LPP defense strategy, the average accuracy loss on the CIFAR-10, GTSRB, and Tiny ImageNet datasets for the ResNet18 model was minimal, with an average drop of only 1.29% in Table 4. This result demonstrates that, despite the scaling and pruning operations involved in our method, the impact on the performance of clean models is negligible, ensuring that the model maintains strong classification performance while defending against backdoor attacks.

Table 4: The impact of LPP defense on the classification accuracy of clean models.

| Dataset | Without LPP | With LPP | Gap |
|---|---|---|---|
| CIFAR-10 | 90.85% | 88.93% | 1.92% |
| GTSRB | 98.56% | 97.56% | 1.00% |
| Tiny ImageNet | 60.69% | 59.74% | 0.95% |

### C.2.1 COMPUTATIONAL EFFICIENCY ANALYSIS

Table 5 shows that LPP markedly outperforms other methods in speed across all datasets, with defence times as low as 0.178 seconds for CIFAR-10. Data-based defences like BNP, I-BAU, and NAD show much higher times, particularly I-BAU, with times exceeding 1500 seconds on the Tiny dataset. This stark contrast in performance underscores LPP's computational efficiency and effectiveness in swiftly mitigating backdoor attacks, positioning it as a highly viable option for real-world applications where rapid response is crucial. In our method, the calculation of the Lipschitz function only requires traversing all network parameters once. Assuming the total number of network parameters is $m$, the time complexity of this computation process is $O(m)$. After calculating the Lipschitz values, performing remove and scale operations, as well as positioning operations, also only involves simple multiplication, hence the time complexity of this part is also $O(m)$. Taking everything into account, the overall time complexity of our algorithm is $O(m)$. We have added this part of time complexity analysis in our new version.

Table 5: Efficiency comparison of various defence mechanisms against backdoor attacks, highlighting the exceptional speed of Lipschitzness Precise Pruning (LPP) across multiple datasets.

| Dataset | Data-based Defence | | | Data-free Defence | |
|---|---|---|---|---|---|
| | BNP | I-BAU | NAD | CLP | LPP |
| Tiny | 38.137 | 1594.2295 | 708.4023 | 0.4179 | 0.2066 |
| GTSRB | 4.304 | 164.2905 | 122.0288 | 0.3361 | 0.1814 |
| CIFAR-10 | 4.2444 | 177.694 | 134.6839 | 0.3633 | 0.178 |

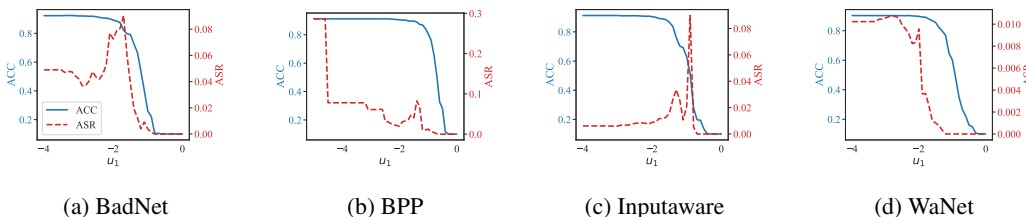

| (a) BadNet | (b) BPP | (c) Inputaware | (d) WaNet |

Figure 4: Performance Variations under Different Lower Bias Limits $u_1$

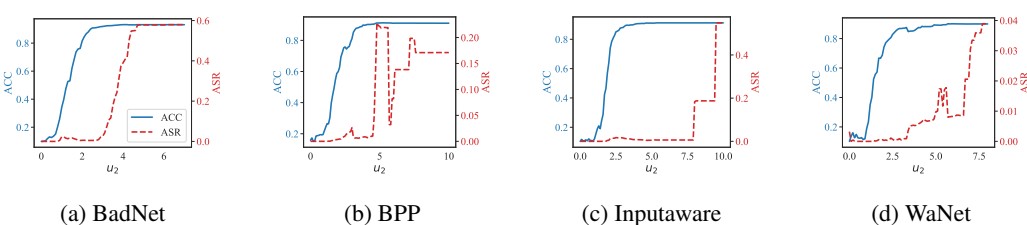

| (a) BadNet | (b) BPP | (c) Inputaware | (d) WaNet |

Figure 5: Performance Variations under Different Upper Bias Limits $u_2$

## C.3 ABLATION STUDY

In this section, we primarily investigate the effects of the upper and lower limits of the degree of bias parameter $u$ in LPP on defence performance, where $u_1$ represents the lower bias limit, and $u_2$ denotes the upper bias limit.

Performance Variations under Different Lower Bias Limits $u_1$: we keep the value of $u_2$ constant and investigate how changes in the lower bias limit $u_1$ affect defence performance. As shown in Figure. 4, it can be observed that with an increase in the lower bias limit, the ACC of models employing the LPP defence method experiences a sharp decline when $u_1$ approaches -1. However, when $u_1 \in [-4, -2]$, the model maintains a relatively high and stable ACC. Moreover, within this interval, a relatively balanced point can be identified, resulting in a generally low ASR for the model.

Performance Variations under Different Upper Bias Limits $u_2$: in this section, we maintain the value of $u_1$ constant and investigate the impact of varying the upper bias limit $u_2$ on the defensive performance of LPP. As depicted in Figure. 5, a similar overall trend is evident. With an increase in the upper bias limit $u_2$, the ACC after applying the LPP defence method gradually increases. Specifically, when $u_2 \in [0, 2]$, ACC experiences rapid growth with the augmentation of $u_2$. Subsequently, within the range $u_2 \in [2, 10]$, ACC stabilizes. In terms of ASR, when $u_2 \in [0, 3]$, it remains at a relatively low level, indicating the robust defensive performance of LPP. However, considering the performance of ACC, when $u_2 \in [2, 3]$, LPP can achieve a high ACC while still having a good level of defence capability.

