# OpenReview forum: "Rethinking Lipschitzness Data-free Backdoor Defense"
_ICLR.cc/2025/Conference — Submitted to ICLR 2025_

### Official Review · Reviewer_zJvG · 2024-10-27

**Soundness:** 2
**Presentation:** 2
**Contribution:** 2
**Rating:** 3
**Confidence:** 4

**Summary:**

The paper addresses the issue of backdoor attacks on Deep Neural Networks (DNNs) and introduces a novel data-free backdoor defense algorithm called Lipschitzness Precise Pruning (LPP). The LPP algorithm leverages the properties of Lipschitz functions to identify and mitigate backdoor vulnerabilities by pruning neurons strongly correlated with backdoor triggers.

**Strengths:**

Key Strengths of the paper include:
1. Proposing a new data-free backdoor defense algorithm that overcomes the limitations of existing methods, which often depend on specialized defense datasets.
2. Optimizing the computation of the Lipschitz constant to efficiently and accurately identify compromised neurons.

**Weaknesses:**

The key weaknesses of the paper can be summarized as follows:

1. **Limited Originality**: The paper's claim to 'rethink' Lipschitzness Data-free Backdoor Defense appears to be a minor adjustment to the existing CLP framework. Specifically, the changes involve substituting the spectral norm with the L2 norm and adjusting parameters close to the pruning threshold. While there is some innovation, the overall enhancement seems minimal.

2. **Insufficient Experiments**: Despite the claim of thorough experimentation, the paper only evaluates and reports results for 4 out of 12 implemented attacks and 4 of 16 defense methods, in the provided code. This raises questions about the completeness and robustness of the evaluation. Additionally, the paper does not compare its approach against state-of-the-art (SOTA) methods such as RNP, which were discussed but not included in the comparative analysis.

3. **Clarity and Writing Issues**: The manuscript suffers from clarity issues that make it difficult to comprehend certain sections. Key problems include but are not limited to:
    - Sections 3.1.2 and 3.2 start with equations without an introduction, which causes confusion to the readers.
    - Equation 4 is confusing, why $f^{(l)}(x)$ has two different outputs?
    - The text from Line 273 to Line 278 is hard to follow, and I cannot understand it when I write this comment.
    - The using of $|$ and $\\|$, as well as the shape of $W_j$ are ambiguous in Line 279-292. Note that $\\|\cdot\\|_2$ has different definitions for matrix and vector.
    -  In Algorithm 1 line 18, should it be "remove $T_{idx}$ parameters in W"?
    -  The blue portion in Figure 2 (Line 338) is confusing as all of them are blue.

    I strongly suggest the authors have some proof-readers to check the notations and statements.

4. **Inconsistencies Between Paper and Code**: There is a discrepancy between the methodology described in the paper and the implementation provided in the code. The paper mentions a single parameter $u$ for pruning, whereas the code uses two parameters $u_1, u_2$. This inconsistency needs to be resolved.

5. **Lack of Experimental Details**: Crucial experimental parameters, such as pruning thresholds $u_1, u_2$, the extreme parameter number $k$, and the bias rate $b$, are omitted.  According to the issues in the CLP Official repo on GitHub, CLP is sensitive to the choice of parameters, and the best choice depends on the model and dataset. Given that similar methods have shown sensitivity to parameter choices, the absence of these details raises concerns about the stability and reproducibility of the results.

6. **Discrepancy in Claims**: The paper suggests that CLP is not suitable for fully connected layers, whereas their proposed method can handle linear layers. However, the proposed method only prunes CNN layers similarly to CLP. Therefore, the stated limitation of CLP might not be a significant issue.

7. **Further Analysis of Originality**: Upon reviewing CLP, it was noted that CLP was initially designed for linear layers and then adapted for CNNs. If fully connected layer are treated as vectors (each slice as a vector), the spectral norm used in CLP reduces to the Euclidean norm used in the new method (LPP). Thus, CLP essentially addresses the same issue with a similar approach to LPP.

It should be noted that I am unable to fully comprehend Issue 2; therefore, its significance has not been validated by me.

**Questions:**

See the weakness, especially 2-6.

---

### Official Review · Reviewer_dPcY · 2024-10-29

**Soundness:** 2
**Presentation:** 1
**Contribution:** 2
**Rating:** 3
**Confidence:** 4

**Summary:**

The paper proposes a novel data-free backdoor defense algorithm called Lipschitzness Precise Pruning (LPP), which leverages the mathematical properties of Lipschitz functions to identify and mitigate backdoor vulnerabilities in DNNs. Traditional defenses require clean data to identify these compromised neurons, which poses limitations due to data availability and privacy concerns. LPP circumvents this by selectively pruning neurons linked to backdoor behavior without needing clean data. The authors evaluate their method on CIFAR-10, GTSRB, and Tiny ImageNet datasets with four baseline defenses under four attacks.

**Strengths:**

1. The topic is of great significance and sufficient interest to ICLR audiences. In particular, data-free backdoor defense is an important research area and worth in-depth exploration.
2. The authors provide the codes of their method to facilitate reproducibility. It should be encouraged.
3. The authors analyze the efficiency of each method. It should be encouraged.

**Weaknesses:**

1. There are some problems with the principle of categorizing defenses.
- Passive defense should also include the detection of backdoored models instead of just detecting poisoned samples
- There are still many other important defense paradigms, such as poison suppression [1]. Please refer to [2, 3] for more details.

2. Missing important references.
- Line 47: no references for active defenses
- Line 56: I believe there are still many other data-free defenses, such as [4]. We can easily find more on [Google Scholar](https://scholar.google.fr/scholar?hl=en&as_sdt=0%2C5&q=data-free%2C+backdoor+defense&btnG=&oq=data)
- Line 125-143: Missing advanced backdoor attacks (e.g., [4-6]). There aren't even any works after 2023. The presentation of existing work is also not systematic.
- Line 144-153: Missing advanced backdoor defenses, such as the SOTA Trigger Inversion [7] and SOTA Input-level Backdoor Detection [8].

3. The contributions of this paper. This is my biggest concern. The authors should explicitly compare their method to CLP and highlight their main contributions.

4.  The authors claim that they revisit the Lipschitz functions. However, the authors failed to highlight their main results and findings.

5. A systematic, structured description of the methodology is missing.

6. No discussion about the resistance to potential adaptive attacks. The proposed method relies on a latent assumption that the backdoor-related features can be decoupled from normal ones. However, adversaries can easily bypass this by reducing poisoning rates, using [4], or designing regularization terms.

7. The results of many important baseline attacks and defenses that I mentioned above are missing.

8. No discussion about potential limitations and future directions.

**References**
1. Backdoor Defense via Decoupling the Training Process
2. Backdoor Learning: A Survey
3. Defenses in Adversarial Machine Learning: A Survey
4. Revisiting the Assumption of Latent Separability for Backdoor Defenses
5. Narcissus: A practical clean-label backdoor attack with limited information
6. Backdoor Attack with Sparse and Invisible Trigger
7. Towards Reliable and Efficient Backdoor Trigger Inversion via Decoupling Benign Features
8. SCALE-UP: An Efficient Black-box Input-level Backdoor Detection via Analyzing Scaled Prediction Consistency

**Questions:**

1. The authors should clarify and highlight their main contributions.
2. The authors should discuss and compare to more (advanced) baseline methods.
3. The authors should discuss the resistance to potential adaptive attacks.

Please refer to 'Weakness' for more details.

---

### Official Review · Reviewer_89NT · 2024-11-02

**Soundness:** 2
**Presentation:** 2
**Contribution:** 2
**Rating:** 3
**Confidence:** 3

**Summary:**

This paper propose a data-free backdoor defense algorithm called Lipschitz Precision Pruning (LPP), which exploits the properties of Lipschitz functions in order to efficiently identify and mitigate backdoor vulnerabilities in DNNs without the need for a clean dataset.  This approach accurately prunes neurons that are strongly correlated with backdoor behavior and adjusts neurons that are weakly correlated. And this pruning strategy can reduce the impact on the overall performance of the network.  The article demonstrates the effectiveness of this approach on backdoor defense compared to existing techniques through experiments on several public datasets.

**Strengths:**

1. Lipschitz Precision Pruning (LPP) allows for the pruning of neurons rather than the entire channel by using the Lipschitz function to compute the Lipschitz constant of a neuron. to provide higher pruning accuracy while preserving the overall integrity of the network.
2. Compared to the Channel Lipschitzness based Pruning (CLP) method, Lipschitz Precision Pruning (LPP) extends the usage scenario of lipschitz constant. It also shows better results than CLP in experiments.

**Weaknesses:**

1. In this paper, although the Lipschitzness Precise Pruning (LPP) method is proposed based on Channel Lipschitznes based Pruning (CLP) with some extensions. However, the core mechanism of this work is still neuronal pruning using Lipschitz constants, which is similar to the core idea of CLP, so it fails to demonstrate significant innovation.
2. Although the article performs a comparison with the CLP, the overall experimental design lacks diversity, especially in the selection of defense models and attack methods used for comparison. In order to highlight the advantages of the LPP method more effectively, more experimental setups inherited from CLP experiment should be introduced for an exhaustive side-by-side comparison.
3. The article did not explore the relationship between the Lipschitz constant and model accuracy (ACC) when performing ablation experiments

**Questions:**

1.  How adaptable is the  Lipschitzness Precise Pruning (LPP) method when applied to various types and sizes of network models?  Specifically, are the results effective when implemented in complex architectures such as Transformers or multi-layered network models?
2.  How should the hyper-parameters involved in the LPP method, such as pruning thresholds, be optimized to achieve the best results?  What impact do these settings have on the overall performance of the model, and how can their optimal values be determined?
3.  How does the LPP method perform against adaptive attacks?  Is it effective in countering adaptive strategies that attackers might use to specifically target LPP’s defense mechanisms?

---

### Official Review · Reviewer_F8xe · 2024-11-04

**Soundness:** 2
**Presentation:** 3
**Contribution:** 2
**Rating:** 5
**Confidence:** 3

**Summary:**

The paper addresses the issue of backdoor attacks on Deep Neural Networks (DNNs), where attackers manipulate models by embedding malicious patterns or "backdoors." Traditional defense mechanisms depend heavily on access to clean, defense-specific datasets, which are often hard to obtain due to privacy and security concerns. To address this limitation, the authors propose a novel, data-free defense method called Lipschitzness Precise Pruning (LPP).

LPP leverages the Lipschitz property to identify and prune neurons associated with backdoor vulnerabilities without needing access to clean data. This approach optimizes the computation of the Lipschitz constant using dot product properties, enabling efficient and accurate detection of compromised neurons. The LPP method prunes backdoor-affected neurons while retaining unaffected ones, which enhances DNN robustness against backdoor attacks.

The method was validated experimentally, showing that LPP can outperform existing data-free methods in protecting DNNs across different datasets without the need for extensive computational resources. This approach has promising applications for real-world scenarios where clean data is scarce or unavailable, providing a practical and effective means to safeguard DNNs against malicious tampering.

**Strengths:**

The paper addresses a meaningful problem by proposing the Lipschitzness Precise Pruning (LPP) method, which tackles backdoor attacks without needing clean datasets. This approach is valuable as it overcomes the existing problem and aligns with current interest in Lipschitz-based method.

The paper is well-structured and clearly written. The authors provide the design and approach, conducting experiments that validate the effectiveness of their method. This strengthens confidence in the LPP approach’s practical utility.

The paper is well-organized, with clear argumentation that makes complex ideas accessible. The authors effectively present the popular Lipschitz-based approach, making it easy to understand how it addresses the problem.

This study provides a practical solution to backdoor attacks, especially in scenarios where clean data is limited, which is highly relevant for security-sensitive applications.

Overall, this paper is well-written and well-structured. It tackles an important topic and provides a meaningful, validated solution, using the popular Lipschitzness-based approach to address real-world issues in DNN security.

**Weaknesses:**

Addressing the data-free issue and introducing Lipschitzness-based Pruning (CLP) were originally done by the CLP paper. While the paper aims to build upon this method, it does not adequately highlight its contributions or novel improvements over the original CLP approach. Explicitly emphasizing how the proposed method optimizes or enhances CLP would better underscore the paper’s contributions. For instance, the authors should elaborate on what specific aspects of the Lipschitzness-based method’s computational costs or processing efficiency they improved, as well as provide more detail on the optimization process and its outcomes.

The enhancement presented in this work appears to be incremental rather than a substantial advancement over the 2022 CLP method. The optimization does not represent a transformative improvement that would justify the novelty of the contribution. To strengthen the paper, the authors should consider reframing their approach to more clearly articulate how it moves beyond CLP in a fundamental way or consider expanding the method’s theoretical grounding to elevate the significance of its contributions.

The experimental evaluation would benefit from more comprehensive and targeted experimentation. For instance, utilizing updated and more representative datasets, backdoor attack scenarios, would make the results more convincing and aligned with current research challenges. Additionally, rather than demonstrating a single performance advantage in specific scenarios, the authors should design a set of experiments that systematically illustrates the superiority of the proposed method over CLP. This could include: A theoretical analysis of the source of advantages, which would form a basis for designing comparative experiments. Comparative evaluations that emphasize and clearly demonstrate the specific performance gaps between this method and CLP on critical metrics.

Relying on only one baseline (the 2022 CLP method) does not sufficiently showcase the method’s relative performance or unique advantages. Expanding the scope of baseline comparisons to include more recent methods would make the results more compelling and help position this work more effectively in the broader context of current research on data-free and pruning methodologies.

**Questions:**

Could you kindly elaborate on the unique contributions of your approach relative to the baseline (CLP)? A more detailed theoretical analysis highlighting specific advantages would make your contributions stand out.

Rather than only presenting a few experiments where the proposed method outperforms the baseline, could you provide a focused comparative analysis? This should include a clear rationale for the method's superior performance in certain metrics, beyond stating that it performs "better."

Perhaps considering updated datasets and diverse attack scenarios could make the experimental results more compelling. Additionally, including more recent baselines in the comparisons would help emphasize your method’s advantages.

---

### Meta-Review · Area_Chair_aUc5 · 2024-12-19

**Metareview:**

This work studied the data-free backdoor defense, and proposed a neuron-level pruning method based on Lipschitz constant.
It received 4 detailed and professional reviews. While most reviewers recognized the importance of this task and the clear writing, the major concern is the limited novelty, since the proposed method is extended based on one existing work called CLP. Both their defense mechanisms and experimental settings are same, without substantial updates on theory or methodology. The authors didn't provide responses. Thus, this work is recommended as reject.

**Additional Comments On Reviewer Discussion:**

This is no rebuttal, and the reviewers' concerns are not addressed.

---

### Decision · Program_Chairs · 2025-01-22

Reject